# Revisiting the Ceara Rise, equatorial Atlantic Ocean: isotope stratigraphy of ODP Leg 154 from 0 to 5 Ma

R. H. Wilkens[1], T. Westerhold[2], A. J. Drury[2], M. Lyle[3], T. Gorgas[4], and J. Tian[5]

[1]Hawaii Institute of Geophysics & Planetology, University of Hawaii, Honolulu, HI, 96822, U.S.A.

[2]MARUM, University of Bremen, Bremen, 28359, Germany

[3]CEOAS, Oregon State University, Corvallis, OR, 97331, U.S.A.

[4]GFZ German Research Centre for Geosciences, 14473 Potsdam, Germany

[5]State Key Laboratory of Marine Geology, Tongji University, Shanghai, 200092, China

*Correspondence to*: Roy H. Wilkens (rwilkens@hawaii.edu)

**Abstract**

Isotope stratigraphy has become the method of choice for investigating both past ocean temperatures and global ice volume. Lisiecki and Raymo (2005) published a stacked record of 57 globally distributed benthic $\delta^{18}O$ records versus age (LR04 stack). In this study LR04 is compared to high resolution records collected at all of the sites drilled during ODP Leg 154 on the Ceara Rise, in the western equatorial Atlantic Ocean. Newly developed software is used to check data splices of the Ceara Rise sites and better align out-of-splice data with in-splice data. Core images recovered from core table photos are depth and age scaled and greatly assist in the data analysis. The entire splices of ODP Sites 925, 926, 927, 928 and 929 were reviewed. Most changes were minor although several were large enough to affect age models based on orbital tuning. A Ceara Rise composite record of benthic $\delta^{18}O$ is out of sync with LR04 between 1.80 and 1.90 Ma, where LR04 exhibits 2 maxima but Ceara Rise data contain only 1. The interval between 4.0 and 4.5 Ma in the Ceara Rise compilation is decidedly different from LR04, reflecting both the low amplitude of the signal over this interval and the limited amount of data available for the LR04 stack. A regional difference in benthic $\delta^{18}O$ of 0.2 ‰ relative to LR04 was found. Independent tuning of Site 926 images and physical property data to the Laskar et al. 2004 orbital solution and integration of available benthic stable isotope data from the Ceara Rise provides a new regional reference section for the equatorial Atlantic covering the last 5 million years.

## 1. Introduction

Sedimentary archives retrieved by ocean drilling since 1968 by the Deep Sea Drilling Program (DSDP, 1968-1983), the Ocean Drilling Program (ODP, 1983-2003), the Integrated Ocean Drilling Program (IODP, 2003-2013) and the International Ocean Discovery Program (IODP, since 2013) provide key records needed to better understand processes and interactions of the Earth system. Over almost 5 decades of coring, ocean drilling samples and data have contributed significantly to major breakthroughs in our understanding of earth history - including such basic tenets as seafloor spreading, a detailed history of reversals of the Earth's magnetic field, evolution/extinction of marine species and many more. Included in this list is the advancement of stable isotope stratigraphy and the recognition of the critical part played by variations in the Earth's orbital parameters in climate history. Sites drilled during ODP Leg 154 on the Ceara Rise have played a significant role in creating age models for the Neogene based on astrochronology.

Stable isotope stratigraphy has become the method of choice for investigating both past ocean temperatures and global ice volume. When global ice volumes are large, such as times of vast continental ice sheets, the world oceans become enriched in $^{18}O$, a "heavy" isotope of the more abundant $^{16}O$. It has been demonstrated (e.g. Hays et al, 1976) that variations in $^{18}O$ enrichment ($\delta^{18}O$) coincide with periodicities of the orbital parameters of eccentricity, obliquity, and precession and their influence on the distribution and intensity of solar insolation on the Earth's surface. Therefore, with a knowledge of the previous behavior of the orbital parameters (e.g. Laskar et al., 2011) isotope stages (cycles) may be assigned ages to a very high degree of precision (astronomical tuning). Lisiecki and Raymo (2005) published a compilation of globally distributed benthic $\delta^{18}O$ records versus age from 57 sites worldwide that included data from the past 5.3 Ma (LR04 stack). Their work established a framework against which almost all subsequent isotopic studies of late Neogene sediments have been compared.

The LR04 stack is a significant contribution for having demonstrated the global semi-synchrony of the overall behavior of the $\delta^{18}O$ record in deep sea benthic stable isotope data. It does, however, have some drawbacks. LR04 is an amalgam of data with various resolutions from sites in different oceans and different latitudes, thus averaging regional signals into the overall stack. The age models used for the individual data sets depend on chronological markers such as the ages of magnetic field reversals that may have changed since the original studies were completed and new data has been reported. Astronomical tuning is complicated by the dominance of obliquity in records from sediments older than 1.2 Ma because the pattern of consecutive cycles are similar. And finally, almost all of the $\delta^{18}O$ profiles were derived from spliced data. Splicing is a technique used at drilling sites to piece together one continuous record from adjacent drill holes (Ruddiman et al, 1987; Hagelberg et al., 1992). Splices may be subject to cycle skipping or duplication of events when data are aligned from different holes. Averaging of multiple sites will compensate for small errors in the spliced records if many sites are used and most have a correct splice. As with age models, splices may evolve over time as more detailed and new types of data are gathered post-cruise and reveal previously missed or doubled $\delta^{18}O$ patterns (see Westerhold et al. 2014, supplementary Fig. S9).

There are 21 records included in LR04 that extend to ages older than 3 Ma included in LR04, and only 14 that have data older than 4 Ma. As the numbers in the stack shrink, the importance of having well-spliced records grows. A number of records used in LR04 contain problematic succession with respect to their composite record.

Site 982, for example, is one of the high-resolution sites that extends beyond 3 Ma (Venz et al. 1999, Venz and
Hodell 2002), and has been used subsequently to transfer age models to other isotope records (Drury et al. 2016).
However, there is controversy over the composite record of 982 as well as the age model (Lawrence et al. 2013,
Khelifi et al. 2012, Bickert et al. 2004).

For the interval 1.7-5.3 Ma the LR04 stack depends heavily on the spliced records from Leg 138 - the S95

benthic composite stack (Shackleton et al., 1995). It was noted in Lisiecki and Raymo (2005) that for marine isotope
stages (MIS) M2 and MG2 at 3.35 Ma there is a mismatch of data and a potential coring or splicing problem in Site
846. Even so, Site 846 was used for the initial alignment in LR04 from 2.7-5.3 Ma along with Site 849 (1.7 - 3.6
Ma) and Site 999 (3.3 - 5.3 Ma). Any problem in the spliced records of the sites used for initial alignment will
propagate through the stack if not compensated by a large number of additional sites. Thus we might expect a
greater possibility of erroneous correlation in older less repeated parts of the stack, particularly where the amplitude
of the $\delta^{18}O$ variations are relatively small (see Lisiecki and Raymo 2005 - Fig. 2).

In order to provide a precise age model the LR04 stack was tuned to a non linear ice volume model forced

by insolation (65°N) using the Laskar et al. (1993) 1,1 orbital solution including an assumed decrease in the lag of
ice sheet response to insolation forcing. To test and evaluate the LR04 stack and the tuning approach from 0 - 5 Ma,
a robust composite record from a single location combined with an astronomical age model that is independent of
ice volume modeling is required. Furthermore, extending the $\delta^{18}O$ stack into the Miocene means that robust
composite records are required to avoid misalignments and tuning errors at the outset. Sediments from the Ceara
Rise (South Atlantic) are perfectly suited for testing because they contain orbitally driven changes and are already
the backbone for astronomical calibration of the Geological Time Scale for the last 14 Ma (Shackleton and
Crowhurst 1997, Zeeden et al. 2013, 2014, Lourens et al. 2004).

Here we revisit data collected during, and subsequent to, ODP Leg 154 (Fig. 1). The LR04 stack includes

benthic stable isotope data from ODP Leg 154 Sites 925, 927, 928, and 929. Site 927 was used for initial alignment
from 0–1.4 Ma.in LR04. Site 926 is also considered a primary site for time scale constructions for 0-15 Ma and is
independent of LR04. In this study, we use newly developed software to check and improve the composite records
of Leg 154. We then stretch and squeeze data outside the splice, use core images to correlate all sites to the Site 926
depth scale, orbitally tune the core images, and compare the age model with the LR04 stack for the past 5 Ma. The
new software system greatly facilitates the construction of benthic $\delta^{18}O$ reference records back into the Miocene
from single regions. Regional astronomically tuned $\delta^{18}O$ records are a next important step in deciphering
paleoceanographic conditions worldwide.

**2. Material and Methods**

The proliferation and diversity of the data collected both during and after ocean drilling cruises can at times

be somewhat overwhelming for the individual scientist. Data are now freely available through online data bases
maintained by the ocean drilling infrastructure for cruise results (e.g. LIMS, JANUS), by national efforts (e.g.
NGDC) or community efforts (e.g. PANGAEA). However, a unified and consistent system for integrating disparate
data streams such as micropaleontology, physical properties, core images, geochemistry, and borehole logging has
not been widely available. In this section we describe a system that we have developed over several years to work
with ocean drilling data and images (CODD - Code for Ocean Drilling Data). CODD takes advantage of the
versatile graphical user interface and analytical functions contained in the IGOR™ graphing and analysis program
commercially available from Wavemetrics, Inc. One of the great advantages of a modern analysis program like
IGOR$^{TM}$ paired with new computers and fast processors is the ability to use images as data. Rather than a static
picture of a core or section, images are scaled and plotted along with traditional data versus depth or age. Core
images may be squeezed, stretched, subsampled, and concatenated, allowing for great versatility. The CODD set of
ocean drilling macros for IGOR$^{TM}$ and a User Guide are freely available at www.CODD-Home.net. Core images,
both as png files and scaled IGOR binaries as well as all tables of this study including age models, offsets, splices,
tie points between sites, spliced MS data, isotope data, and mapping pairs for squeezing and stretching of cores are
available through the open access Pangaea website under https://doi.pangaea.de/10.1594/PANGAEA.870873.

**2.1 *Data Structure***
The heart of the CODD data structure is the coring matrix - a 3 layered array in which the top layer
contains the original depth to the top of each section (mbsf - meters below seafloor) sorted by core (rows) and
sections (columns). The middle layer contains the length of the sections and the third layer the composite depth
(mcd - meters composite depth). Sample depths are calculated by referencing the proper layer and coordinate by
core and section and then adding the sample interval. The reverse process of returning the core, section, and interval
designation of a given sample depth is accommodated by comparing it to the section top depth plus the section
length to find where the sample originated.
A standardized naming convention is essential to efficient processing of multiple and diverse data streams.
In CODD data are assigned 3 part names, Hole, Technique and Information, separated by underscores. Thus
gamma-ray attenuation depths are U925A_GRA_MBSF and U925A_GRA_MCD with data as U925A_GRA_GRA.
Core, section, interval and age are similarly named. Isotope data might be U925A_Iso_d18O and U925A_Iso_d13C.
While the Hole and Technique designations must be identical for a single data set, the Information may be anything
the user desires, including new data like ratios created from existing information. IGOR$^{TM}$ records data processing
steps and the use of a standard naming convention allows users to repeat processing for different data by simply
replacing one Hole or Technique with another in the recorded steps. It also simplifies the development of
automation macros. This is essential for processing large amounts of data from multiple drill-holes and drill-sites -
especially when changes to composite records (splices) are needed.

**2.2 *Image Processing***
Ever since IODP Leg 200, core section images have been captured by line scanners as discrete files which
are easily loaded into analysis programs with little or no preparation. However, the only access to core images from
the first approximately 200 ocean drilling cruises are through digitized photographs of entire cores laid out on a
table in parallel sections (Fig. 2A). CODD includes a module for cutting core section images from core table photo
images, correcting them for uneven lighting, scaling them to mbsf (meters below seafloor) and combining them into
a single core image (Fig. 2B) through a series of simple steps. In general, the outer 5% of each section image is
excluded to minimize friction effects of coring that tend to bend horizontal layers. In practice it takes between 1 and
2 minutes to go from loading a core table photo to producing a scaled composite core image. The visualization and
impact of the scaled composite is very much different from the core table photo and of much greater value during
data analysis. The use of scaled composite core images has proven to be particularly effective in creating site splices
or for the checking of existing splices.

Lighting correction is a necessary step when using images cut from core table photos because the light

source used for the original photos was co-located with the camera above the center of the core table, resulting in the
center of the picture being brighter than the edges (Schaaf and Thurow, 1994; Nederbragt and Thurow, 2001, 2005).
This effect is illustrated by profiles of lightness from H S L (hue, saturation, lightness) representations of section
images plotted together (Fig. 2A inset). For these sections the variability of the intensity of lightness, excepting
some spikes representing darker layers, is around 50 units of lightness (out of a full scale of 0 - 255). The difference
from the center to the ends of the best-fit line to the profiles is approximately 25 lightness units, so uneven lighting
has a significant effect on the section images. When the core table photos are viewed, the observer's eyes and mind
make a correction and the uneven lighting seems subtle, but we have found that when stringing section images
together to make a composite core image the 1.5 m long lighter/darker cycles are readily apparent. As many ocean
drilling sediment cores vary in lightness as a function of carbonate and/or biogenic silica content (e.g. Balsam et al.,
1999), lighting cycles in core images degrade the usefulness of core color or lightness profiles as proxies for other
properties of interest or for spectral analysis. Thus CODD processing of core table photos includes a step which fits
a line to the lightness profiles and then applies a "flattening" filter which brightens the section images away from the
center according to the fit. While not perfect, the process removes most of the 1.5 m color cyclicity (Fig. 2B). There
is also lighting variation across the core box images that can produce a 9/10 m cycle in the spliced composite
images. It appears to be somewhat more diffuse than the along-core section variation and hasn't hindered the present
work. We are developing a process to correct for lighting variation of the entire core box image prior to cutting the
individual section images. This may also allow us to remove the color cast present in many of the older core box
images, such as the purplish hue seen in Fig. 2A.

**2.3 *Splicing, Stretching, and Squeezing***

In the same manner that sections may be strung together to make a composite core image, extracted splice

sections of core images from different holes can be merged into a single scaled spliced site image (Fig. 3a). Splicing
is a 2 step process, the first of which involves offsetting the mbsf depth for individual cores to a composite depth by
aligning features in data collected from multiple holes. It is worth noting here that it is rare that all features in
individual cores from different holes align - as coring disturbance (e.g. extension or compression at the top and
bottom of piston cores, see Ruddiman et al, 1987 for an in depth discussion) or natural variability mean that while
one feature may align, another is offset (e.g. Lisiecki and Herbert, 2007). The individual setting the splice (the
correlator) makes a decision as to which feature to align based on overall considerations of the splicing process.
Once the core offsets are set, the correlator chooses tie points between holes to produce as complete a sedimentary
record as possible while avoiding any possible duplication. In the past this has been done using data profiles of
properties measured on whole round core sections - primarily density from Gamma Ray attenuation (GRA), and
magnetic susceptibility (MS) as well as reflectance spectrophotometer intensity (RSC) on split sections. This can
prove to be tricky when using data that are replete with similar cycles. Cycle skipping or doubling is a constant
source of potential error and the inclusion of images in the process helps greatly. While checking splices or splicing
cores and choosing tie points we used the same criteria as typically used by the shipboard stratigraphic correlator for
(I)ODP expeditions. The splice should contain no coring gaps and disturbed sections are avoided. Where possible
we avoided using the top and bottom ~0.5 m of cores, where disturbance resulting from drilling artifacts is most
likely. Those portions of the recovered core most representative of the overall stratigraphic section of the site are
picked and the number of tie points is minimized to simplify sampling.
An example from Ceara Rise Site 927 demonstrates image utility while examining an existing splice. A 10
m long section of images and data is presented in Fig. 3. Poor agreement between offset data from all three holes of
Site 927 occurs around 50 mcd, immediately below a splice tie in the published splice for the site (Fig. 3A). The
images show poor agreement between the light and dark bands in cores 927C-05H and 927B-06H. A better solution
is obtained by reducing the offset of 927B-06H by 1.6 m to align the peak in RSC seen around 50.2 mcd in 927C-
05H with a similar peak at 51.8 mcd in 927B-06H (Fig. 3B, 3C). Fortunately, because the core images are depth
scaled, CODD allows us to shift and re-splice both core images and all other datasets using a simple algorithm. The
resultant shift shows better agreement between images and data from both holes. Significantly, the shift illustrated
removes one 40 ka obliquity cycle from the isotope record (Bickert et al. 1997) and will alter a tuned age model
accordingly.
Traditionally, once the splice has been set, subsequent samples are taken and measurements made only
from the core material included in the splice. While three or more holes are often cored at sites devoted to
paleoceanographic studies, the volume of samples available within a splice is equivalent to a single hole. And since
archival halves of each core are reserved for later sampling, it is often difficult to obtain new samples along a
heavily sampled section of the splice. More material is available from sections of cores not included in the splice,
but as mentioned above, the process of aligning and offsetting cores from adjacent holes by matching features is
imperfect due to coring effects and natural variability (e.g. Lisiecki and Herbert 2007, Wilkens et al, 2009).
Misalignment of off-splice features may add significant noise when in-splice and out-of-splice data are combined. In
order to align features from sections of core not included in the splice it is necessary to stretch/squeeze images and
data outside the splice. Magnetic susceptibility data have been stretched from the off-splice data to the splice in Fig.
4. Using CODD, sets of tie points between off-splice data and the splice for each hole (yellow numbers in Fig. 4) are
selected using cursors. Stretched data and images are updated in real time. The tie points allow investigators to
interpolate out-of-splice mcd depths to their equivalent levels in the splice.
The ability to squeeze and stretch data and images has a second useful application. Sites drilled in the same
general area of the ocean, such as those on the Ceara Rise, often share many physical features in data such as
density, magnetic susceptibility, or color in their sediment columns. In a manner similar to the process of stretching
and squeezing off-splice data to the splice, CODD employs a cursor driven routine to stretch data and images from

different sites to a single common depth scale using similar features. The segment of the stretch of Site 927 to Site 926 between tie points 60 and 80 is illustrated in Fig. 5. In total, 428 pairs of tie points were identified while matching the upper 304 mcd of Site 927 to the upper 285 mcd of Site 926. Additional constraints such as paleomagnetic reversals and biostratigraphic events may be included, helping to guide the correlation. In practice a user views multiple data types and images simultaneously and tie points selected from one data set are mapped to all others at the same time.

**2.4 *Depth to Age***

Once data and images from the individual sites have been tied to a common depth scale the final CODD processing step is to set everything to a single age model. We used the age models of Bickert et al (1997) and Tiedemann and Franz (1997), adjusted for our splice corrections and updated to Laskar et al. (2004), to compare age-scaled images and data from the various Ceara Rise sites. An example comparing Sites 926 and 927 is presented in Fig. 6. Comparison of the composite images is remarkable for the fact that individual sedimentary layers that represent sometimes less than 10 kyr are readily identifiable between sites. This suggests that in areas where the sediment has enough color variation highly targeted samples may be collected that represent precisely the same event at multiple sites.

MS data and the composite image of Site 926 are compared with orbital calculations using Laskar et al. (2004) in Fig. 7. The orbital curve was calculated using 100% of the eccentricity (E) effect plus 50% of the obliquity (T) and precession (P) intensities. Correlation of the MS data to the Laskar model was the primary basis for the Bickert et al. (1997) and Tiedemann and Franz (1997) age models, so agreement between the 2 curves is expected. They used a correspondence between MS maxima and northern hemisphere summer insolation minima to develop their age models. This phase relationship was found to be most consistent in both precession and obliquity frequency bands (Shackleton and Crowhurst 1997). See Zeeden et al., 2013 for a concise description of their approach. Comparison with a composite core image was not possible for those earlier investigators and our results illustrate the remarkably detailed agreement between cycles seen in the calculations and variations in sediment color. Based on these observations and the well-known phase relationship (Bickert et al. 1997) we refined the tuning for Site 926 tying dark (light) layers, which correspond to MS maxima, to ET-P minima (maxima). We used only the core image and color reflectance for tuning; therefore plotting the magnetic susceptibility data versus insolation serves as a crosscheck for a consistent phase relationship throughout the record.

**3. Results**

We checked the entire splices of Sites 925, 926, 927, 928 and 929 for the last 5 Ma. Most of the changes in the published splice tables were minor although several, such as the one illustrated in Fig. 3, were large enough to affect age models based on orbital tuning. Data from samples outside of the revised splices were aligned with the splice based on stretching and squeezing of the out-of-splice data. Mapping pairs to convert depths outside of the splice to the composite depth are provided in supplemental files. For the interval spanning 0 to 5 Ma we compiled 5533 benthic $\delta^{18}O$ isotope measurements from Bickert et al. (1997), deMenocal et al. (1997), Tiedemann and Franz

(1997), Shackleton and Hall (1997), Billups et al (1998) and Tiedemann and Franz (1997). Data were plotted on the
updated age model for Site 926. Data from all of the sites are compared with one another and a smoothed curve
(Gaussian filter) combining all of the sites is compared to LR04 in Fig. 8. Data tables for core offsets, splices, and
age models are available as supplemental files to this publication.
Agreement amongst the different Ceara Rise Sites is good in terms of the shapes of the curves while there
is a spread in absolute values. This is likely due to the water depths at the different sites, which ranged from 3040 m
at Site 925 to 4355 m at Site 929. Offsets in benthic oxygen isotope data between Site 925 and Site 929 in some
intervals (e.g. 3.6 to 4.5 Ma) have been suggested to indicate a relatively warmer and saltier NADW than today
(Billups et al. 1997).
The overall agreement between the Ceara Rise smoothed composite oxygen isotope curve and the LR04
global compilation is generally quite good although there is a definite difference in absolute values with the Ceara
Rise data exhibiting consistently lower values of about 0.2 ‰ than LR04 (Supplementary Fig. S1). The 0.2 ‰ offset
is well within the potential regional differences of up to 0.3 ‰ cited by Lisiecki and Raymo (2005). The consistency
of the difference over the entire 5 Myr scope of this study is remarkable given the regional mix of data used for
LR04.
While the agreement between Ceara Rise and LR04 oxygen isotope data is good, there are discrepancies in
some intervals. The 2 curves are out of sync between 1.80 and 1.90 Ma with LR04 exhibiting 2 maxima whereas
Ceara Rise contains only 1. As this is close to a point where the LR04 stack switched from Site 677 (0-2 Ma) and
Site 927 (0-1.7 Ma) to Site 849 (1.7-3.6 Ma), misalignments in the stack between single sites with the original
spliced records could have led to a mismatch here. Tuning for Site 926 in this interval is robust and does not allow a
shift that could accommodate the mismatch. Hence the interval from 1.80 and 1.90 Ma in the LR04 stack has to be
revised. Even larger differences are seen between 4.0 and 4.5 Ma (Fig. 9). Data from Site 929 have been shifted
+0.25 ‰ in Fig. 9 to aid in the comparison of the excursions in the data. The data from Sites 925 and 929 are in
good agreement, but the Ceara Rise smoothed compilation, which is almost entirely composed of data from the 2
sites over this age interval, bears little resemblance to LR04. As pointed out in Lisiecki and Raymo (2005), their
stack prior to 4 Ma includes far fewer sites than the more recent data. The 4.0 to 4.5 Ma interval is also one of low
amplitude variability in $\delta^{18}O$ as a response to orbital variation, making the tuning effort at the individual sites
contributing to LR04 more difficult than at later time intervals. Better correlation of data older than 4.5 Ma suggests
that age model uncertainties are confined to 4.0 - 4.5 Ma and do not necessarily offset the age models for older
sediments in LR04 or our compilation.
Accessing uncertainty in the age model is difficult and cannot be discussed in this manuscript as it would
require extensive testing. However, in Zeeden et al. (2013) and (2014) this is already done with regards to the
uncertainty in the target curve. The outstanding match of sedimentary pattern and insolation calculations, which is
amazing, keeping in mind that the Laskar et al. 2004 model is based on a relatively short time of observational data,
gives confidence that the error for the Miocene is less than a single precession cycle. Due to the excellent match in
patterns we think the main error lies in the accuracy of the target (precession and obliquity). The error in precession
maxima and minima positions will be only relevant for times older than 5 Ma (see Lourens et al. 2004), as already
discussed in the Zeeden et al. (2013, 2014) papers.

**5. Discussion**

Independent tuning of Site 926 images and physical property data to the Laskar 2004 orbital solution and

integration of available benthic stable isotope data from the Ceara Rise provides a new regional reference section for
the equatorial Atlantic covering the last 5 million years. Comparing the CODD based new stack from the Ceara Rise
to the LR04 stack reveals overall very good agreement suggesting that most of the LR04 stack is robust for the
interval from 0-4 Ma. Disagreement in the interval from 1.8-1.9 Ma (Fig. 9) points to uncertainties in the records of
Sites 677 and 849. The record of Site 677 (Shackleton et al., 1990) has a gap in the composite around this time
interval at 85 mcd. Our unpublished re-examination of the Mix et al. (1995) Site 849 age model suggests that it
might be affected by issues in the composite record revolving around core 849C 5H at around 52 mcd. Construction
of an equatorial Pacific stack, presently underway, should resolve the issue.

The differences between LR04 and the Ceara Rise average between 4 and 4.5 Ma reveals a more complex

matter that questions assumptions made in LR04. The tuning in Site 926 (Fig. 10) in this interval is robust and can
not be changed. The match between the precession-dominated insolation curve and the dark/light pattern shown in
the composite site image is excellent. To match the LR04 and the Ceara Rise isotope stacks, the Ceara Rise stack
needs to be shifted by 21 kyr to older ages between 4.1 and 4.3 Ma - which is not possible without changing the
phase relation between insolation and the dark/light pattern of the Ceara Rise sediments. The LR04 stack is basically
tuned to obliquity in this interval with lighter $\delta^{18}$O in obliquity maxima. The major discrepancy at 4.2 Ma occurs in
an interval of low obliquity amplitude and higher precession amplitude modulation (Fig. 11). Lighter $\delta^{18}$O values
match insolation maxima in the interval around 4.2 Ma, thus suggesting that the cyclic changes in $\delta^{18}$O are related to
precession rather than obliquity. Moreover, the minimum in $\delta^{18}$O at 4.18 Ma and the maximum at 4.21 Ma in the
Ceara Rise stack do not correlate to obliquity minima and maxima as they do before and after this interval, which
coincides with a minimum in the 1.2 myr obliquity amplitude modulation. A closer look at the individual isotope
records at Ceara Rise (Fig. 12) reveals that these cycles are indeed precession cycles, seen in the site composite
image as well as in the benthic $\delta^{18}$O data. We therefore conclude that the LR04 stack misinterpreted these two cycles
as one obliquity cycle that then was used to tune the LR04 age model. According to the Ceara Rise tuning this
interval is not related to obliquity but rather to precession variations. This means that the assumption in LR04
matching all cycles to obliquity is dangerous in intervals of low obliquity amplitude and can lead to incorrect tuning
results.

Further study of splices and age models used in the data contributing to LR04 will be needed before these

discrepancies can be fully resolved. Such clarification is a necessary step in the ongoing effort to create a global
correlation of isotope and other data that can be resolved at the isotopic stage level. Such examination of other areas
of the oceans will also aid in the development of regional isotope curves to compare with our findings for the Ceara
Rise. The CODD approach is a useful tool for extending oxygen isotope reference records into the Miocene and
beyond. Combining multiple records from several sites drilled in an oceanic region is greatly facilitated by CODD
and helps to form a regional stratigraphic framework. Stacked records from different regions, such as the equatorial
Pacific, are urgently needed to test and verify the completeness of each record as gaps can occur on a regional scale.
Establishing high resolution age models on a regional scale is  key to understanding paleoceanographic changes on
orbital timescales for the entire Cenozoic.

**6. Conclusions**
We have demonstrated a new system for capturing core images as data using newly developed CODD
software. The ability to transform core table photos and line-scans of core sections into data as depth or age scaled
core images has helped greatly in the task of revising published splices for Ceara Rise sediments cored during ODP
Leg 154. Comparison of the revised data with the LR04 global oxygen isotope stack reveals that there are sections
of the stack that are not well resolved. Further study of data contributing to LR04 will lead to a clarification of the
misfits we have found as well as establishing other regional isotope offsets from a global stack. The CODD software
package thus can play a key role in the construction of a new generation of the benthic isotope stack and surely will
be very helpful in extending the stack into the Miocene. The next important step will be to form a more robust and
accurately tuned initial signal used to form the benthic isotope stack.

**Acknowledgements**
Development of CODD was partially supported by post cruise funds from U.S. Science Support for RW.
Financial support for this research was also provided by the Deutsche Forschungsgemeinschaft (DFG) to TW and
AJD.

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

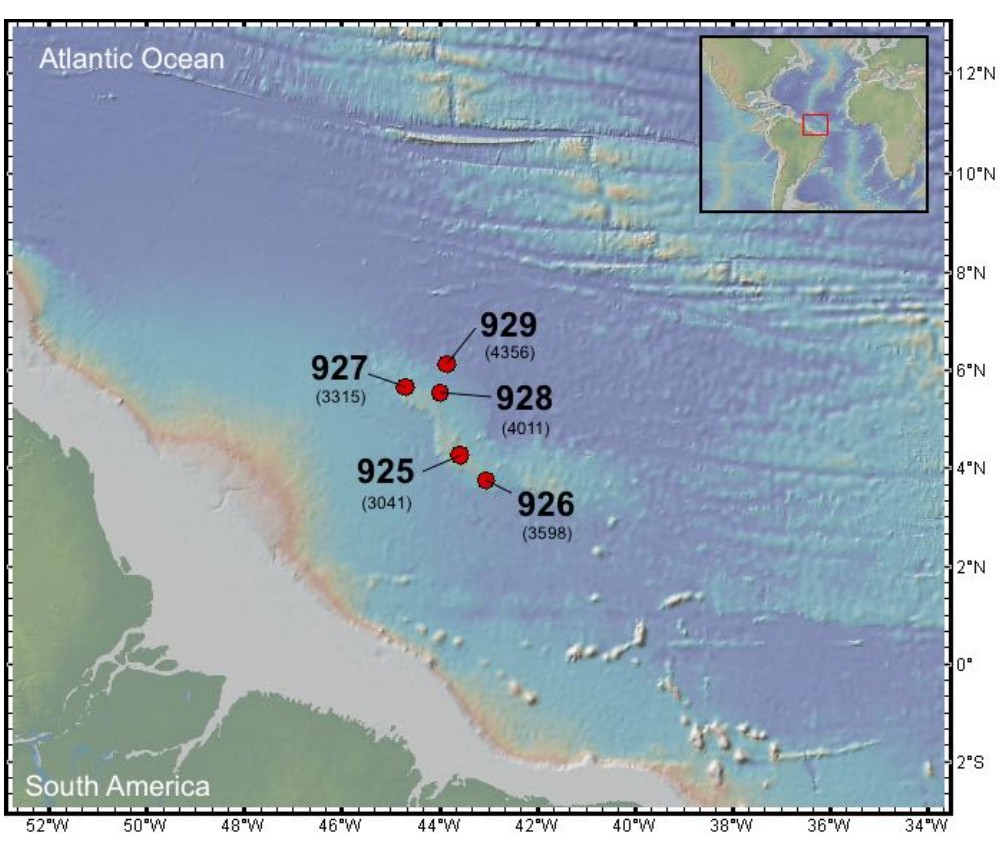



Figure 1: The location of ODP Leg 154 Sites.


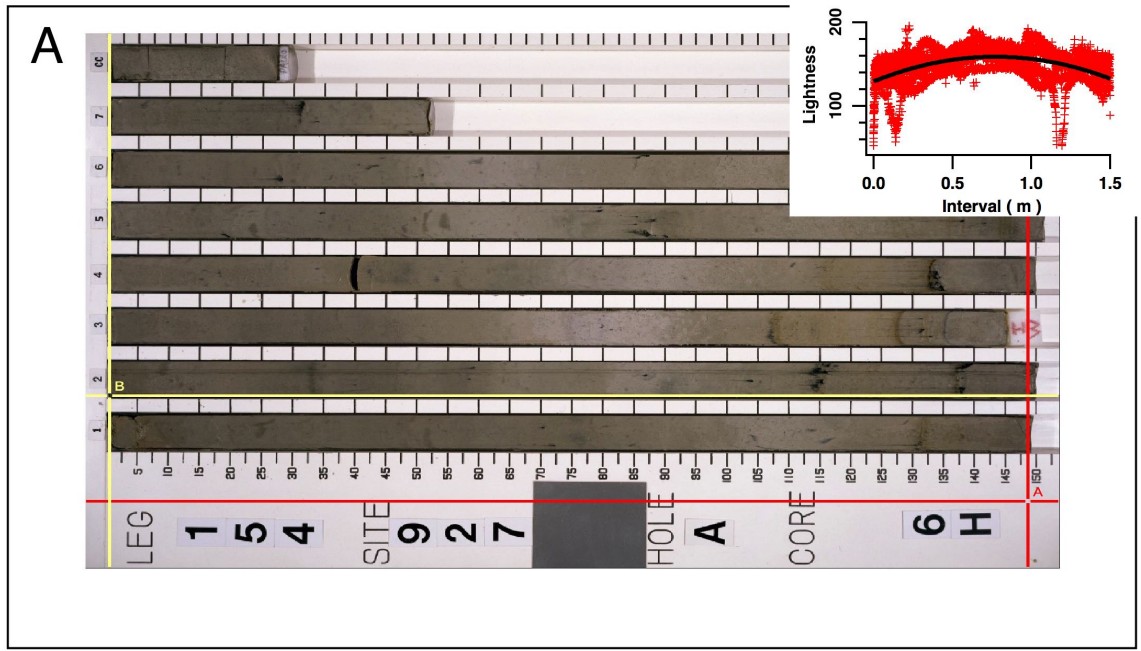

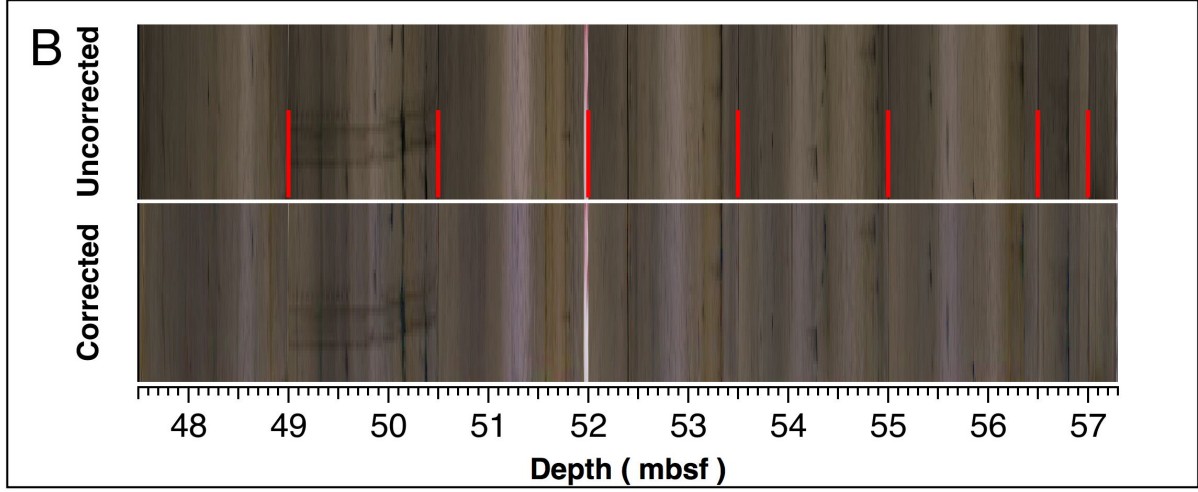


Figure 2: Creating a composite core image from a core table image. (A) Image loaded into IGOR. Red cursor moves horizontally
to set bottom locations in pixels of each section. Yellow cursor moves horizontally and vertically to the lower left corner of each
section before cutting. Inset - Lengthwise lightness profiles for each of the cut sections and a best fit line used for the lighting
correction. (B) Composite core image scaled to mbsf. Vertical red lines indicate section breaks. Lower image has been corrected
for uneven lighting in the core box photo.

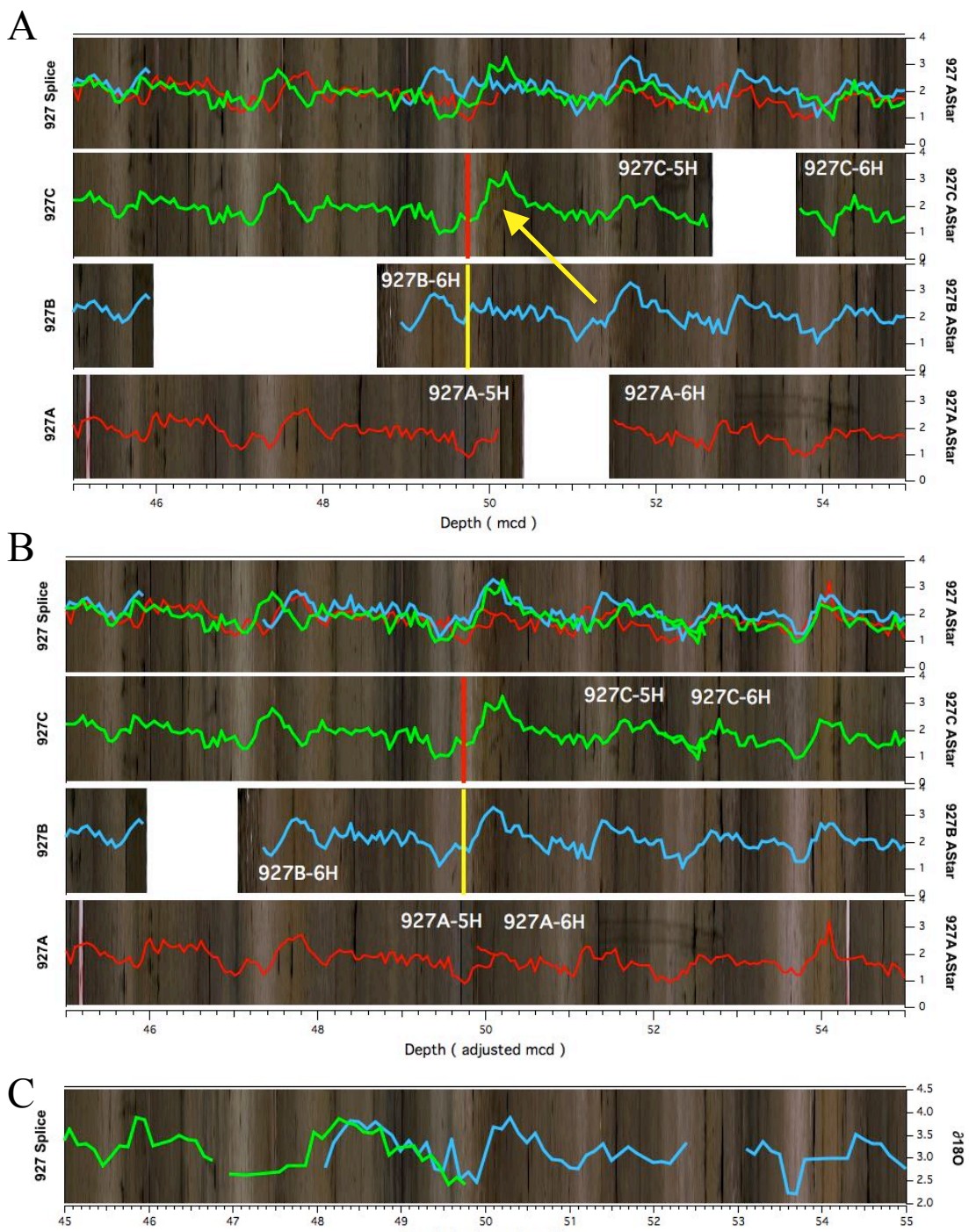



**Figure 3: A.** Reflectance spectrophotometer (RSC) a* data (LAB color model) and core images plotted against the published splice mcd. The yellow arrow indicates misaligned features. The yellow vertical line represents the top of a splice section and the vertical red line shows the bottom of the previous splice section. **B.** The revised splice. The splice goes from Core 927C-05H to Core 927B-06H in both cases, but the offset for Core 927B-06H has been reduced by 1.6 m in the revised splice to account for the repeat sampling of a cycle. Note the poor agreement of the data between 49 and 51 mbsf in the original splice. **C.** Benthic δ18O revised. Samples were collected based on the original splice, resulting in data duplication between 48 and 50 m adjusted mcd.

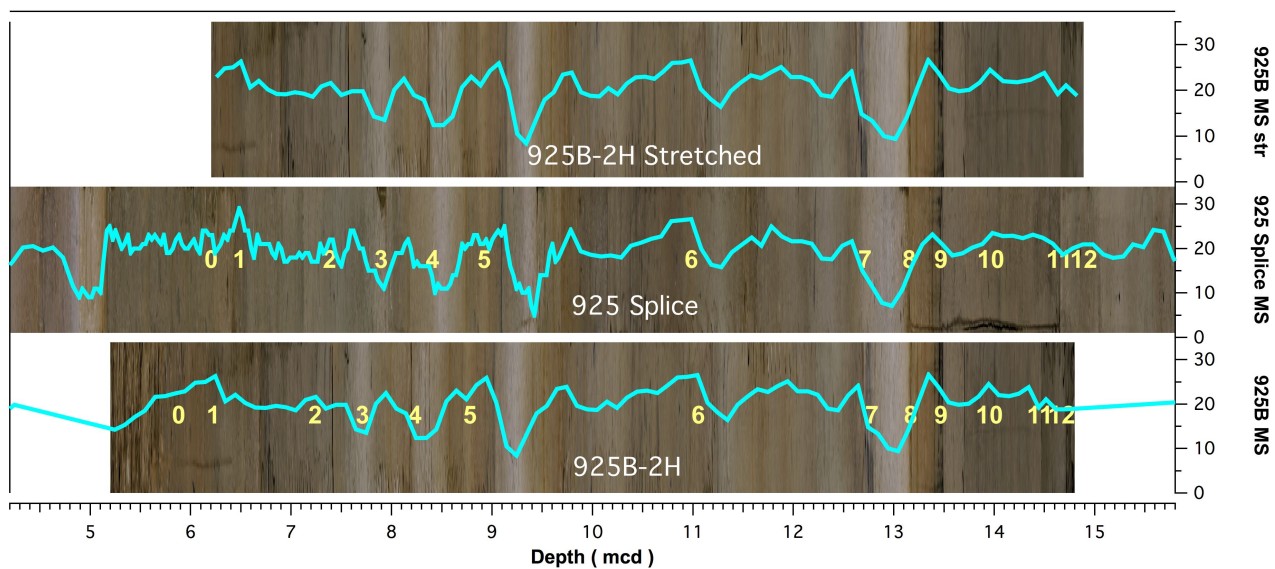

**Figure 4: Core 925B-2H was not used for the  Site 925 splice and while there is good alignment between the core image and data and the spliced image and data at 13-14 mcd, shallower portions of the core are not well aligned with the splice. Yellow numbers indicate tie points used to stretch the image and data so that they are in better agreement with the splice. Choice of tie points is cursor driven and stretching can be recalculated in real time.**

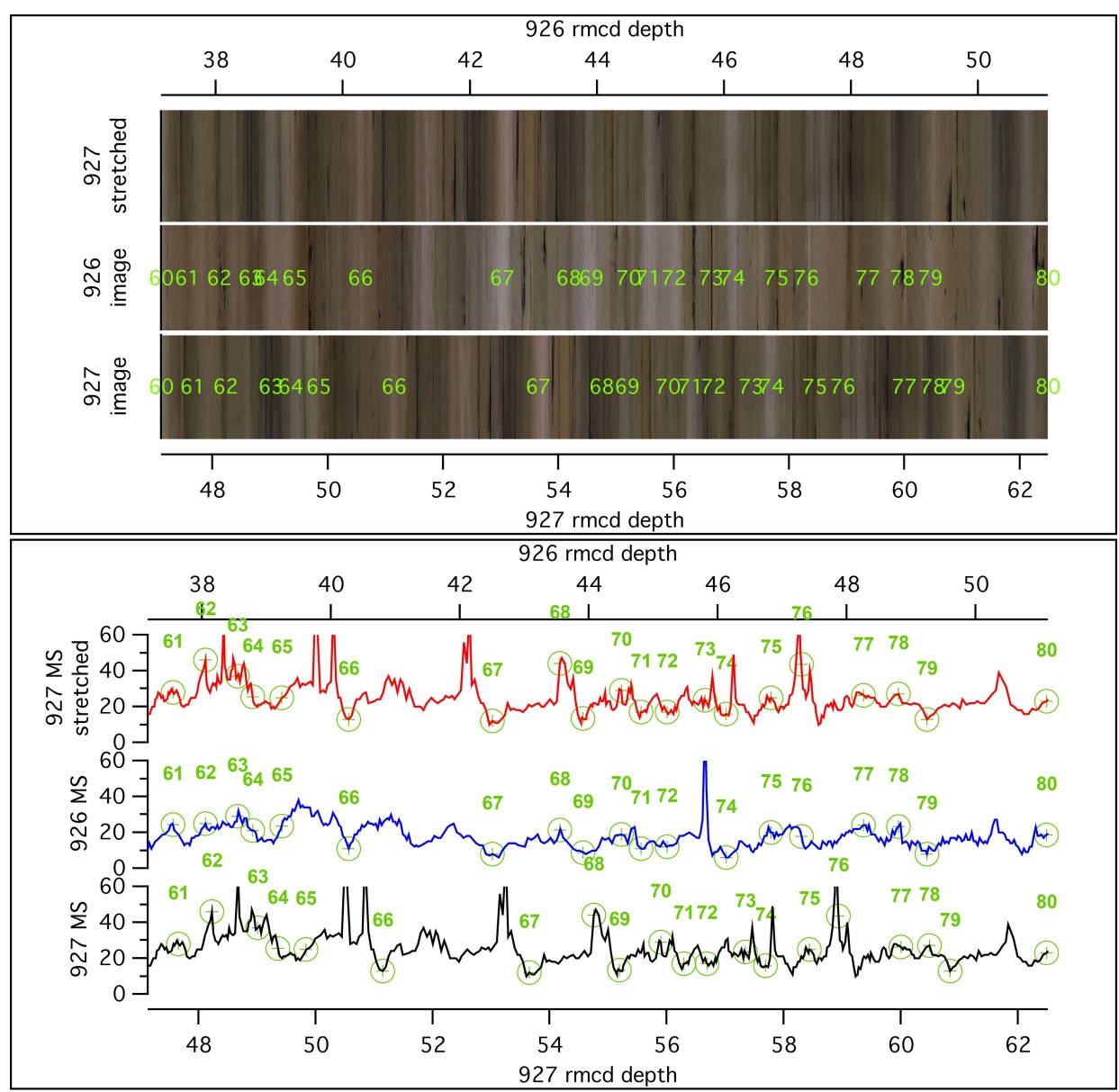

**Figure 5: Spliced images and MS data from ODP Sites 926 and 927. The rmcd depth scales indicate that there have been small adjustments to the published splices for each site. Site 927 data and image are plotted versus the Site 927 depth scale on the bottom of each graph and versus the Site 926 depth scale at the top. Green numbers indicate tie points between the sites used to stretch the Site 927 image and data.**

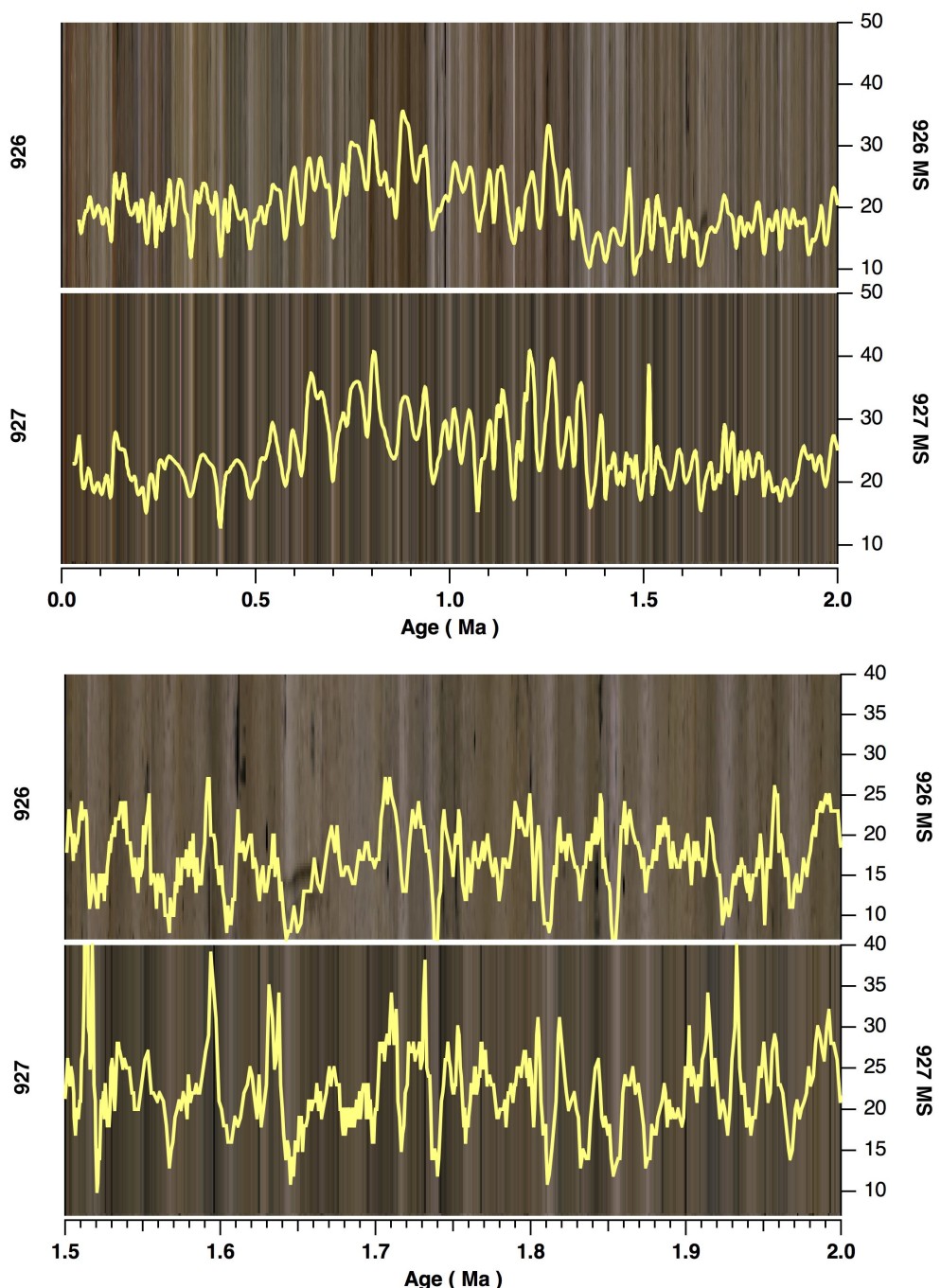


**Figure 6: Top - Smoothed MS data and images plotted versus age from 0 - 2 Ma. Bottom - 1.5 - 2 Ma detail using non-**
**smoothed data. Fine layers, on the order of 10 kyr, are correlated between Sites 927 and 926.**

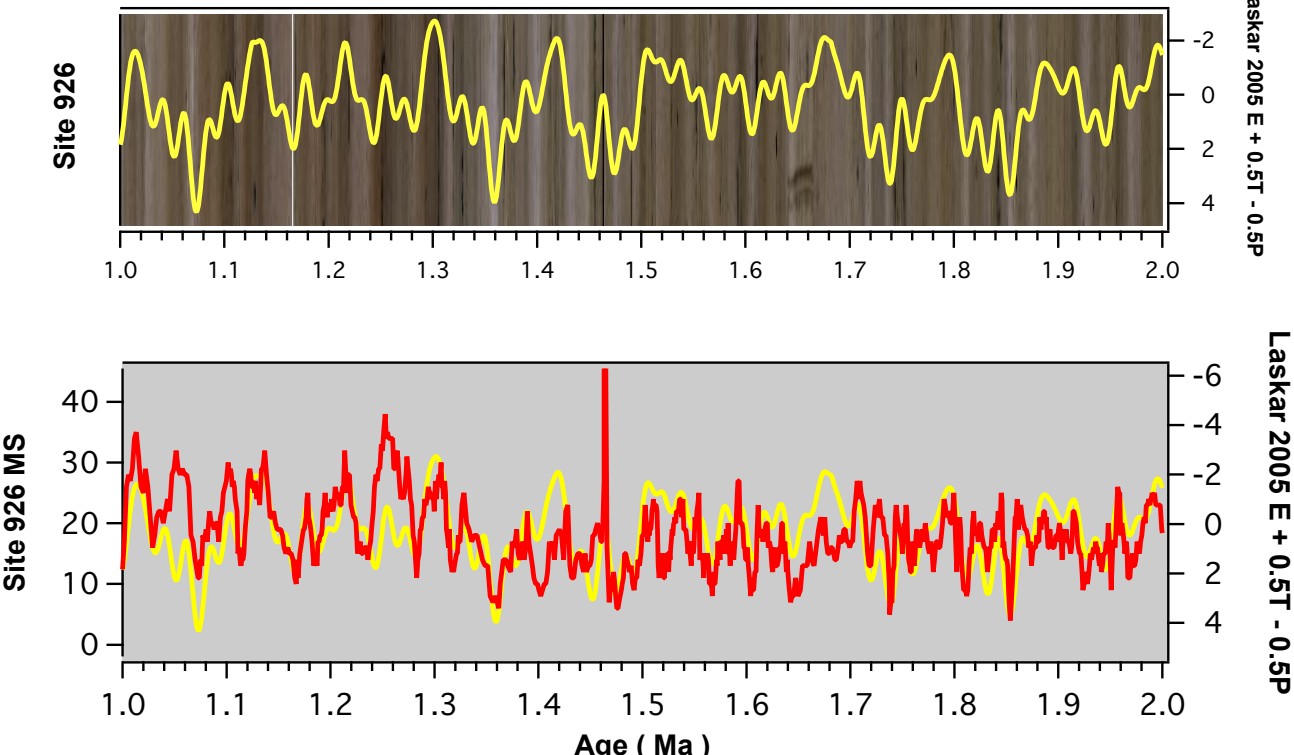


**Figure 7: Laskar et al. (2004) orbital calculation compared to the Site 926 composite image and MS data. E = eccentricity,**
**T = tilt ( obliquity ), and P = precession. The Laskar curve was compared to MS to check the age model used in this study**
**that was based on the images and color reflectance. The composite image is the result of comparing multiple data sets and**
**individual core images.**


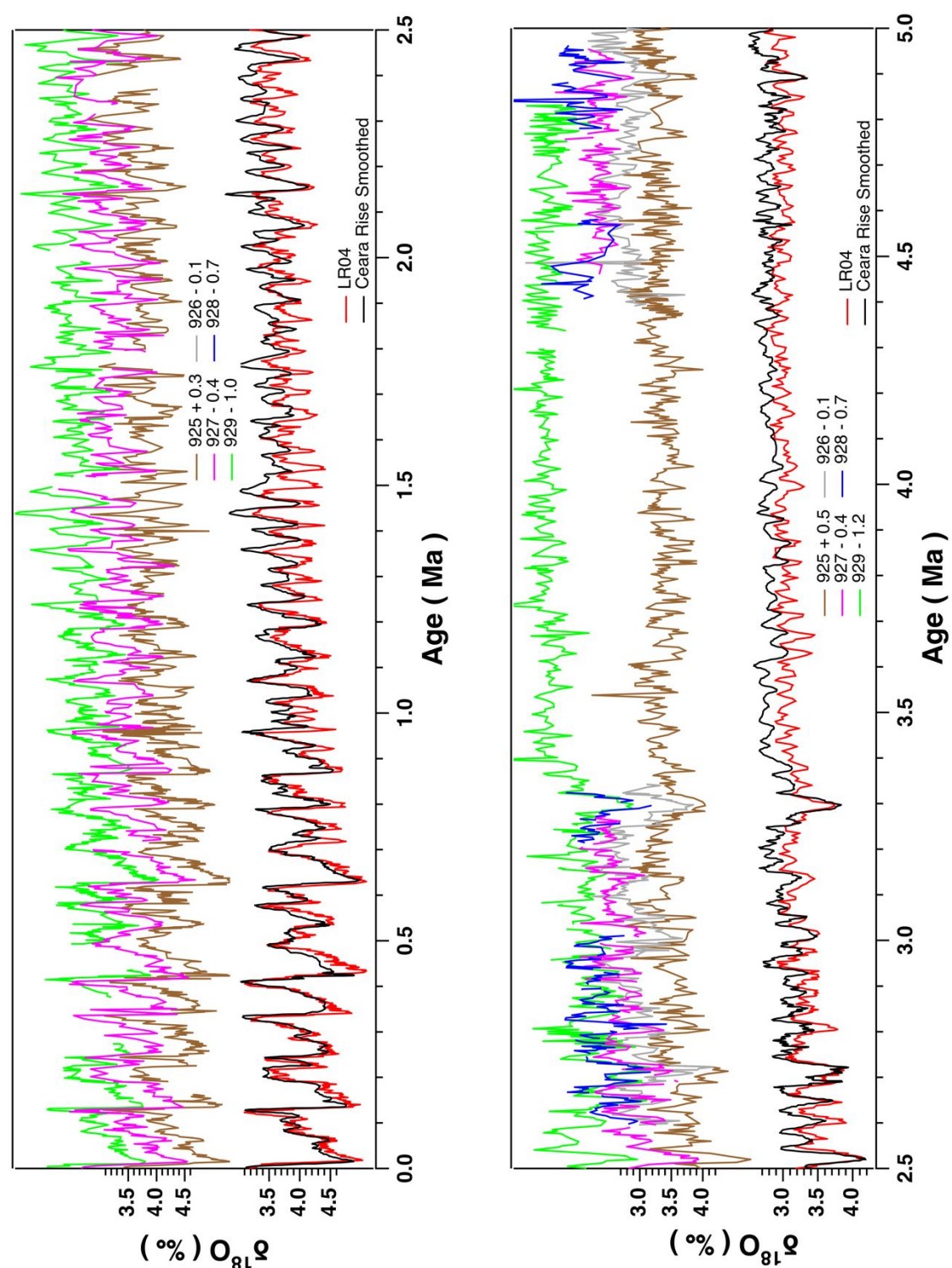


**Figure 8: Benthic oxygen isotope data from all Ceara Rise sites compared with one another and a smoothed composite of**
**all data compared to LR04. Top - 0 to 2.5 Ma, bottom 2.5 to 5 Ma. Note the δ¹⁸O scale change between top and bottom**
**plots. Indvidual site traces have been offset as indicated in the legend.**
Original Figure 9 was removed.

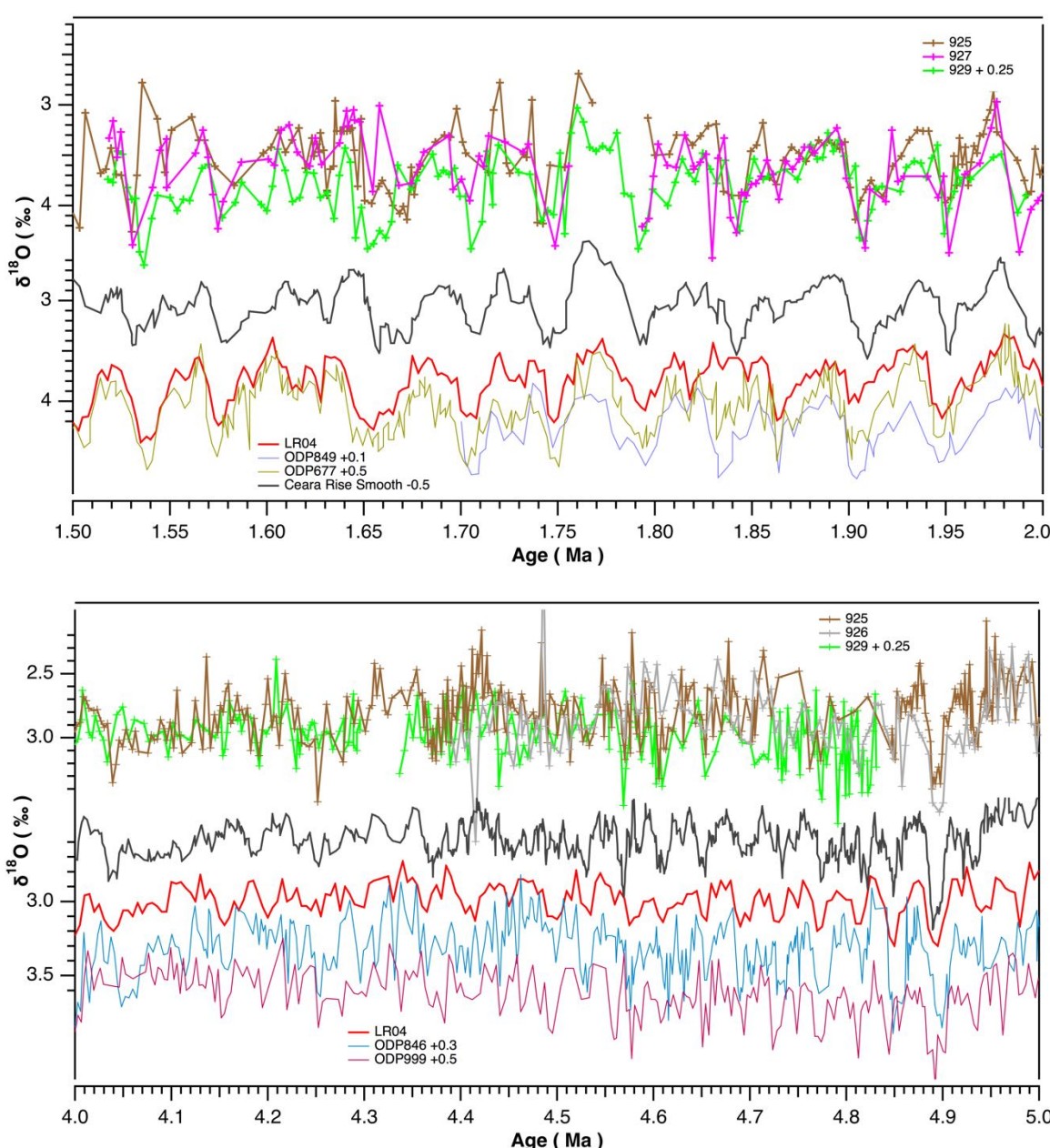


**Figure 9: Detail from Fig. 8 comparing individual holes to one another and a smoothed composite to LR04 for the intervals 1.5 to 2.0 Ma and 4.0 to 5.0 Ma. For better illustration we plotted the initial alignment target records of the LR04 stack. For the 1.5 to 2.0 Ma interval these are the records from ODP Sites 677 and 849, for the interval 4.0 to 5.0 Ma these are the records from ODP Sites 846 and 999. Some records have been shifted as indicated in the figure for better comparison of the data with each other. Note the differences between LR04 and the Ceara Rise average at 1.80 - 1.85 Ma although the initial alignment targets are more similar to the Ceara Rise smooth. Also note the difference between 4.0 and 4.5 Ma. The Site 999 record is from a single hole and the splice of the Site 846 record might be erroneous. The age model for the Ceara Rise is very robust in this interval (see. Fig. 10) pointing to potential inconsistencies in the age model construction of the Site 846 and Site 999 records.**

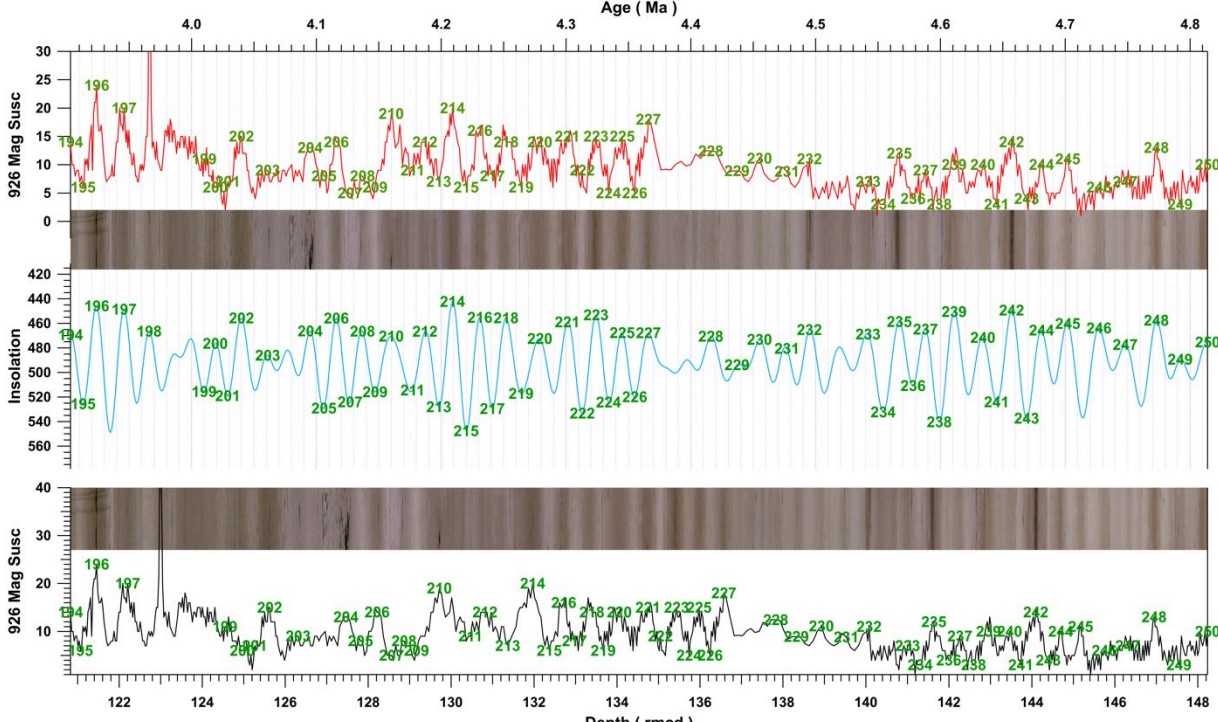

**Figure 10: Detail from CODD tuning of Site 926 magnetic susceptibility and core images to insolation. Bottom is data versus depth, middle shows insolation 65°N 21st June inverted, and top shows image and magnetic susceptibility versus tuned age. Green numbers mark position of tie points. Numbers identify tie points between the data and the insolation curve. Light/dark layering in the composite core image is tied to precession cycles prominent in the insolation curve.**

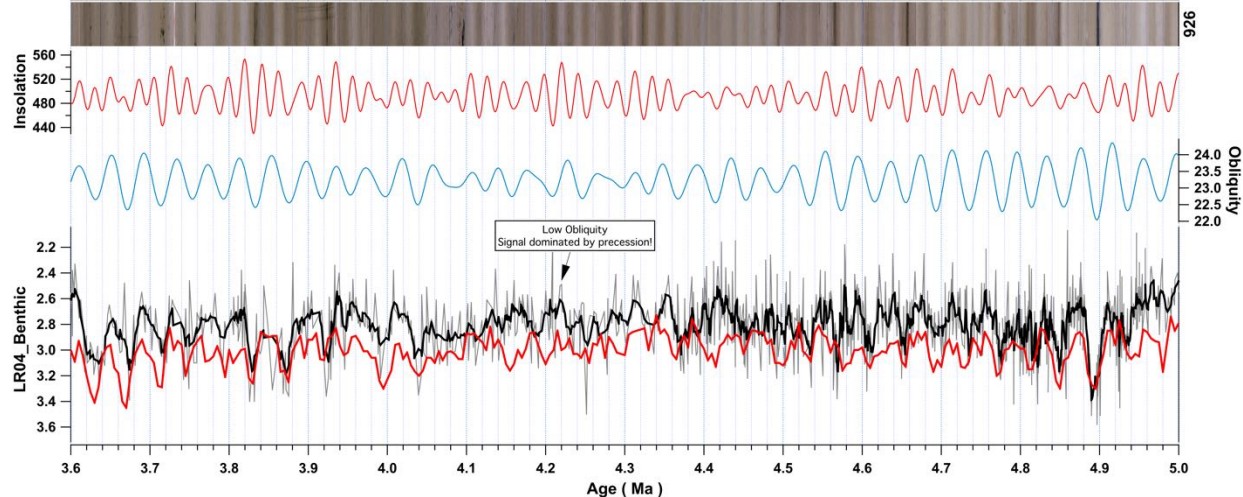


**Figure 11: A comparison of LR04 (Red) to Ceara Rise (grey and black (smooth)) to obliquity and insolation from Laskar**
**et al. 2004. Note that the interval 4.0 and 4.5 Ma exhibits poorly defined obliquity cycles leaving insolation dominated by**
**precession.**

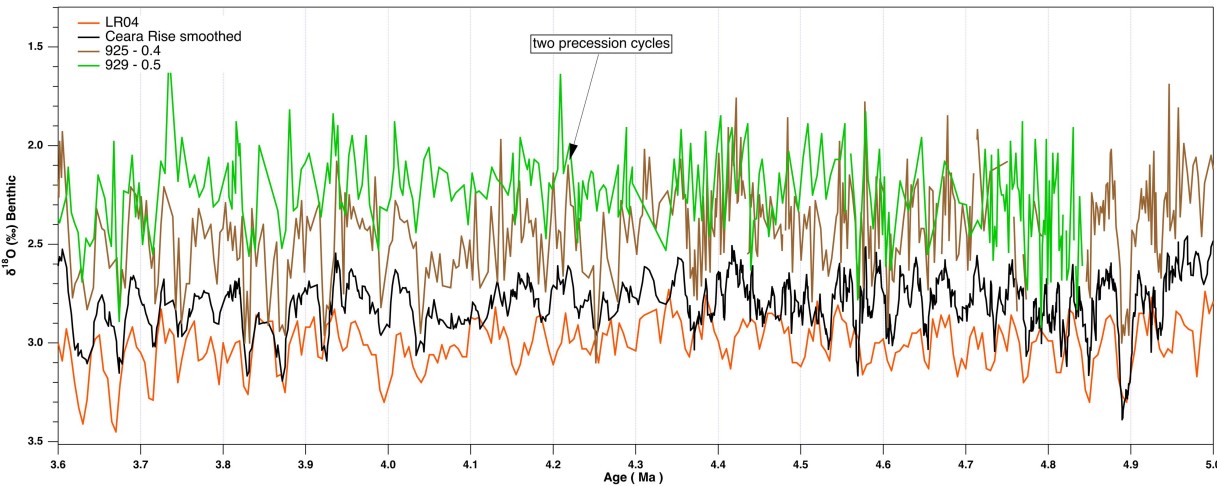



**Figure 12: A comparison of LR04 and Ceara Rise (smooth) to Site 925 and Site 929 benthic isotope data. LR04**
**assignment of variability in the interval from 4.0 to 4.5 Ma to precession peaks may have resulted in the mismatch with**
**the Ceara Rise stack.**

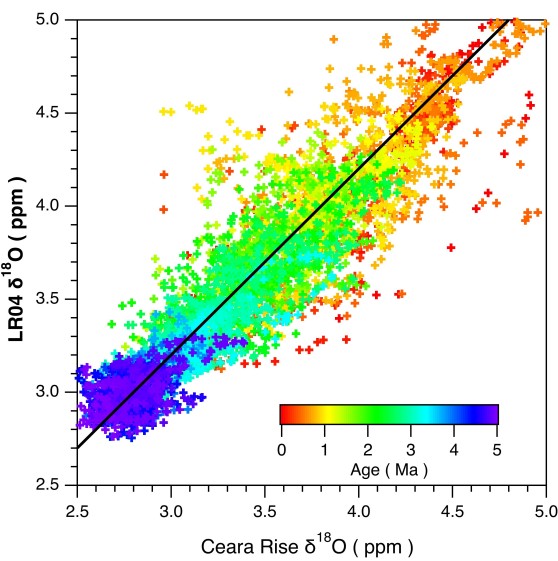

**Figure S1: A comparison between the oxygen isotope data from the smoothed Ceara Rise composite and the LR04 global compilation. The black line represents a 1:1 correspondence that has been shifted by +0.2 ppm along the LR04 axis.**


IGOR CODD Functions, a User Guide, and Help files may be downloaded at www.CODD-Home.net.

Data Tables may be found on the PANGAEA database at https://doi.pangaea.de/10.1594/PANGAEA.870873.
The tables include:

For each site

Offset table

Splice interval table

Spliced magnetic susceptibility ( MS ) data including Site 926 equivalent depths

Isotope data including Site 926 equivalent depths

Age model including Site 926 equivalent depths

Stretching tie points for each hole ( offsplice depths vs splice depths )

Table of species abbreviations for isotope tables

Leg 154 Combined benthic isotope data

Leg 154 Smoothed benthic isotope data

Site to site tie tables linking sites 925, 927, 928, and 929 to site 926

Core images ( lighting corrected ) for all Leg 154 cores in png format with depth scale and as depth scaled

546                    Igor binary files