# Peer review of "Revisiting the Ceara Rise, equatorial Atlantic Ocean: isotope stratigraphy of ODP Leg 154 from 0 to 5 Ma"

_Climate of the Past, 2016_

## Referee Comment (RC1) · C. Zeeden (Referee) · 3 Feb 2017

Dear Authors,

With great interest I have read your manuscript entitled 'Revisiting the Ceara Rise, equatorial Atlantic Ocean: isotope stratigraphy of ODP Leg 154'. In this manuscript you describe the CODD software package, which supplements the commercially available IGOR software system, and use this software to splice all ODL Leg 154 Sites for the last 5 million years. In addition, you extract color (Lab) information from 'old style' ODP core images and correct these for the light source. Finally you align all available d18O data of the Ceara Rise record to the Site 926 splice and create a smoothed composite record, and compare this Ceara Rise composite to the LR04 benthic isotope stack (Lisiecki and Raymo, 2005).

The compilation of an equatorial Atlantic benthic isotope stack, its comparison to the established LR04 stack, and the at least in some intervals higher quality of the new stack is a substantial advance in paleceanography and paleoclimate research. The limitation to the last 5 Ma may go into the manuscript title.

The overall quality of your manuscript is without doubt high, figures are explaining the content in detail and are showing your data handling coming to the final dataset. Your manuscript clearly is a substantial contribution to Paleoceanographic data; your compilation of the Ceara Rise isotope stack is important for the wide readership of Climte of the Past, and relevant for a wide paleocimatic and wide paleoceanographic community. Generally, you describe the data handling in detail, and tie points used are supplied in Supplementary Materials, which make your approach reproducible. You use established methods of data generation, splicing and tuning to come to the age model and data set for the Ceara Rise isotope record. You consider earlier and similar work in an appropriate way. More specific points are given below. Figures supplement the text in a logical way, and Supplementary Materials provide additional data.

Generally I think clarifying following points would be beneficial for your manuscript

You provide a final dataset which is smoothed. Also because not all software that you use is open source, I would suggest to consider also providing the full isotope dataset on the spliced record. This will facilitate other researchers to determine uncertainty, and provides the option to use other smoothing methods. In addition, I suggest to supply the magnetic susceptibility record of the splice to allow for a detailed investigation of the tuning. Could you also comment on the possibility of using this record for alternative age models?

Generally, you make reference to the original tuning of the record (Bickert et al., 1997; Tiedemann and Franz, 1997) and the well-known phase relationship. In my opinion repeating their approach in several more sentences here would be beneficial for many readers.

Chapter 2.2.: If I understand this correct, you do not correct the ODP files for horizontal differences in colour? This is not necessarily a problem here, but should be clarified. In addition, this would imply that ca. 9/10 m cycles in colour may be related to this effect – please mention this.

Similar experiments of extracting data and correcting for the light source have been made by (Nederbragt and Thurow, 2001, 2005) and I suggest to give reference their work here.

In lines 269/269 you mention that a more robust age model will be helpful, if I understand this correct. This stands in contrast to previous suggestions of a known phase relationship. In my opinion a clarification, and an estimation of age model accuracy would be helpful for a wide readership, also to assess similarity of pattern, age and age uncertainty reported by (Lisiecki and Raymo, 2005).

You use quite some figures, some of which may not be very relevant for a wide readership. I suggest considering to move few figures to supplements.

-

In addition, I would like to mention several minor points at specific places:

Line 33: I suggest to add a sentence on the value of the Ceara Rise record of ODP Leg 154

Lines 43(ff): 'synchrony': I would rather suggest to use semi-synchrony or a similar expression

Line 56f: 'that may have shifted' – I suggest to express more clear that their reference age may have shifted due to the availability of new records

Line 49: Splicing is commonly used in ocean drilling, but not limited to 'Ocean' – I suggest to remove 'ocean' here.

Line 50: cyle → cycle

Line 97ff: please explain all abbreviations to include the wide range of readers of Climate of the Past

Line 126: Probably you mean units of lightness – please clarify the meaning of numbers here a bit clearer.

In chapter 2.3. you discuss core disturbance, I also regard (Ruddiman et al., 1987) a good reference here – please decide yourself to include this or not.

In chapter 2.3. you refer to Fig. 3a, but not 3b and 3c – this may be useful in the manuscript text.

Line 179: . . . to a common depth scale?

Line 194: include 'for the last 5 Ma'?

Line 251f: Here it may be stated that high eccentricity intervals may be expected to lead to such patterns – but this is not necessary, please decide yourself.

297: Reference – there seems to be an issue with special characters.

Figures: Adding 'Site' to Site numbers may be helpful for non-(I)ODP involved readers

Fig 6: you use 'ka' in the heading and 'kyr' in the text

Fig. 7: Labelling of x-axes may be added.

Fig. 9: The crosses seem integer numbers, while the legend suggests a smooth transition of colours

Line 405: 'ans' should read 'and'?

Line 406: 'darl' should read 'dark'?

Fig. 12: isotope data seem to miss an axes and/or labelling.

-

References:

Bickert, T., Curry, W.B., Wefer, G., 1997. Late Pliocene to Holocene (2.6-0 Ma) western equatorial Atlantic deep-water circulation: Inferences from benthic stable isotopes, in: Proceedings of the Ocean Drilling Program. Scientific Results. Ocean Drilling Program, pp. 239–253.

Lisiecki, L.E., Raymo, M.E., 2005. A Pliocene-Pleistocene stack of 57 globally distributed benthic $\delta$18O records. Paleoceanography 20, PA1003. doi:10.1029/2004PA001071

Nederbragt, A.J., Thurow, J.W., 2005. Digital Sediment Colour Analysis as a Method to Obtain High Resolution Climate Proxy Records, in: Francus, P. (Ed.), Image Analysis, Sediments and Paleoenvironments, Developments in Paleoenvironmental Research. Springer Netherlands, pp. 105–124.

Nederbragt, A.J., Thurow, J.W., 2001. A 6000 yr varve record of Holocene climate in Saanich Inlet, British Columbia, from digital sediment colour analysis of ODP Leg 169S cores. Mar. Geol. 174, 95–110. doi:10.1016/S0025-3227(00)00144-4

Ruddiman, W.F., Cameron, D., Clement, B.M., 1987. Sediment Disturbance and Correlation of Offset Holes Drilled with the Hydraulic Piston Corer - Leg 94. Initial Rep. Deep Sea Drill. Proj. 94, 615–634.

Tiedemann, R., Franz, S.O., 1997. Deep water cicrculation, chemistry, and terrigenous sediment supply in the equatorial Atlantic during the Pliocene, 3.3-2.6 Ma and 5-4.5 Ma. in: Proceedings of the Ocean Drilling Program. Scientific Results. Ocean Drilling Program, pp. 299–318.

---

## Referee Comment (RC2) · T. Bickert (Referee) · 13 Feb 2017

Dear Authors,

Christian Zeeden already did a very careful review of your manuscript, so please find only some addtional remarks from my side. In general, I fully agree with Christian that your revision of the Plio-/Pleistocene benthic oxygen isotope stratigraphy of Lisiecki and Raymo (2005) using a revised stack of records in the small region of Ceara Rise is a substantial and highly appreciated benefit for paleoceanographic research. It was a pleasure to read the manuscript, and to my opinion it is clearly worth being published in Climate of the Past. I have only a few comments which might help to improve the contribution.

Lines 125ff: Tremendous work that has been done to digitalize the older core pho-

tographs, and to correct them for uneven brightness. However, due to wall friction during core penetration, many sections show parabolic bent layers. Is that a problem, when splicing the images of different holes, which might be affected differently by friction? Furthermore, these images are compared to data sets, which have been measured just in the center of the splitted cores (e.g., magnetic susceptibility using a point sensor, reflectance photospectrometry). Is the offset to a core-wide integration of image data of any importance?

Lines 140ff: For readers not that familiar with the splicing procedure, I would suggest to explain in more detail the criteria, how the "master record" is chosen out of the aligned holes. This is in particular important, to understand, why in a second step the sections outside the splice may be streched and squeezed, instead of being implemented in the master record with their original length.

Lines 215ff: Since the discrepancy between of the interval 1,80 to 1,90 Ma is the largest in the Pleistocene part of the LR04, you should maybe illustrate what might have been the problem for Lisiecki and Raymo (2005) in that interval by exhibting the original records used (see also suggestions for Fig 10).

Lines 223ff: I fully agree that the tuning of the distinct cyclicity in lithology to orbital precession is robust and of good help as a control for oxygen isotope stratigraphy in the interval between 4.0 to 4.5 Ma. However, again I would prefer to see in separate figure, what might have been the problems of LR04 tuning in that interval, to better follow the arguments presented in the discussion (lines 236ff).

Fig 7 Abbreviations on 2. y-axes should be explained in the caption. Laskar 2005 => 2004?

Fig 8 Since this figure contains the main results of the study, I would suggest to stretch the two graphs and present them on one page each in a portrait format. Furthermore, the offset of the individual d18O records should be raised to better get access their correlation. Larger data gaps (in particular in sites 927 and 929) should be left open.

Fig 10 I would re-organize this figure in a way that below the results of the Ceara Rise stack, you should probably present at the original data sets of the LR04 stack, to get behind the problem of the former stratigraphy within the interval 1.8 to 1.9 Ma. The lower graph should be moved into a separate figure, and maybe stretched to better present the details of the interval 4.0 to 5.0 Ma.

---

## Author Comment (AC1) · 28 Feb 2017

Dear Christian Zeeden,

Thank you very much for your very constructive review.

Here we will reply directly to your review.

*The compilation of an equatorial Atlantic benthic isotope stack, its comparison to the established LR04 stack, and the at least in some intervals higher quality of the new stack is a substantial advance in paleoceanography and paleoclimate research. The limitation to the last 5 Ma may go into the manuscript title.*

We will revise the title to "**Revisiting the Ceara Rise, equatorial Atlantic Ocean: isotope stratigraphy of ODP Leg 154 from 0 to 5 Ma**". In fact, the splices of all sites were corrected as far down core as possible. Including rechecking the age model this covers the interval back to 15 Ma. In this paper we focus on the last 5 million years because most of the isotope data are published from this interval and it allows comparison of our results to the LR04 stack.

*You provide a final dataset which is smoothed. Also because not all software that you use is open source, I would suggest to consider also providing the full isotope dataset on the spliced record. This will facilitate other researchers to determine uncertainty, and provides the option to use other smoothing methods. In addition, I suggest to supply the magnetic susceptibility record of the splice to allow for a detailed investigation of the tuning. Could you also comment on the possibility of using this record for alternative age models?*

We will add a table with the full isotope data compilation together with the smoothed isotope record to the dataset and mention this in the revised manuscript. In addition we will add the spliced magnetic susceptibility, color reflectance, and GRA density spliced for each site.

We think the pattern of the magnetic susceptibility records can be tuned as done in your Zeeden et al. 2013 paper. Here we wanted to be independent from this approach and only use images and color reflectance data. Subsequently, plotting the magnetic susceptibility data versus insolation provides a crosscheck for a consistent phase relationship throughout the record. We will clarify this point in the revised version.

*Generally, you make reference to the original tuning of the record (Bickert et al., 1997; Tiedemann and Franz, 1997) and the well-known phase relationship. In my opinion repeating their approach in several more sentences here would be beneficial for many readers.*

The phase relation for tuning is discussed in detail in your Zeeden et al. 2013 paper as well, chapter 5.2. We will add a mention of the phase relationship between insolation and MS as well as add the 2013 citation to the text.

*Chapter 2.2.: If I understand this correct, you do not correct the ODP files for horizontal differences in colour? This is not necessarily a problem here, but should be clarified. In addition, this would imply that ca. 9/10 m cycles in colour may be related to this effect – please mention this.*

*Similar experiments of extracting data and correcting for the light source have been made by (Nederbragt and Thurow, 2001, 2005) and I suggest to give reference their work here.*

You are correct concerning the ~10m cycles that may appear in the lighting-corrected image data. We will point this out in the text. We have been working with spectral analysis of image lightness profiles versus depth and there is in fact a peak at 0.1 cycle/m that is relatively constant despite changing sedimentation rates down hole. True cycles, such as precession, shift position down hole with varying sedimentation rate.

Lighting correction is a complicated issue. You can see in Figure 2 that there is a purplish color cast to the digitized photo as well as uneven lighting. We are now working on a macro for lighting & color correction of the entire core box image that we would like to apply prior to cutting the core sections from the image. Ideally this will remove most of the 10m effect.

We were not aware of the work by Nederbragt and others and appreciate the references which will be added to the text.

*In lines 269/269 you mention that a more robust age model will be helpful, if I understand this correct. This stands in contrast to previous suggestions of a known phase relationship. In my opinion a clarification, and an estimation of age model accuracy would be helpful for a wide readership, also to assess similarity of pattern, age and age uncertainty reported by (Lisiecki and Raymo, 2005).*

The passage you mention is: "The CODD software package thus could play a key role in the construction of a new generation of the benthic isotope stack and surely will be very helpful in extending the stack into the Miocene. The next important step will be to form a more robust and accurately tuned initial signal used to form the benthic isotope stack."

This sentence refers to the other records used for the LR04 stack as initial references. From the Lisiecki and Raymo 2005 paper: "Our initial alignment targets are high-resolution segments of seven d18O records: GeoB1041 from 0– 0.15 Ma, ODP Site 1012 from 0 – 0.6 Ma, ODP Site 927 from 0–1.4 Ma, ODP Site 677 from 0–2.0 Ma, ODP Site 849 from 1.7 – 3.6 Ma, ODP Site 846 from 2.7 – 5.3 Ma, and ODP Site 999 from 3.3 – 5.3 Ma". Basically for refining the LR04 stack in the interval older than about 4 Ma the Leg 138 records (Site 846, 849) need to be splice-checked. The data from Site 999 are from a single hole only (999A), and NOT a splice. Given our experience splicing other sites it is possible that as much as 10% of the sediment column of Site 999 is missing. Data from these 3 sites need to be correlated to the revised records now presented here from Ceara Rise and their age models modified accordingly. This effort is already in progress.

Accessing the real uncertainty in the age model is difficult and cannot be discussed in this manuscript as it would require a lot of testing. However, in your Zeeden et al. 2013 and 2014 papers this is already done with regards to the uncertainty in the target curve. The outstanding match of sedimentary pattern and insolation calculations, which is amazing

keeping in mind that the Laskar et al. 2004 model is based on a relatively short time of observational data, gives confidence that the error for the Miocene is less than a single precession cycle. Due to the overwhelmingly good match in patterns we think the main error will be lying in the accuracy of the target (precession and obliquity). The error in precession maxima and minima positions will be only relevant to times older than 5 Ma (see Lourens et al. 2004 in the GTS book), and this is already discussed in the Zeeden et al. 2013, 2014 papers.

With respect to the exact ages for terminations and interglacials of the Quaternary we refrain from doing so here, because this is not the scope of the manuscript. The overall match to the LR04 stack from 0-3 Ma shows that the straight forward tuning as done in the manuscript is reliable.

*You use quite some figures, some of which may not be very relevant for a wide readership. I suggest considering to move few figures to supplements.*

One of the main goals of the manuscript is to introduce the new CODD macro software package. We move step by step to allow the reader to understand how the CODD software functions and is used to construct the Ceara Rise compilation. The CODD software is a visually driven tool and thus we would like to keep all the figures to illustrate its functionality. We will move figure 9 to the supplement ( the comparison between the oxygen isotope data from the smoothed Ceara Rise composite and the LR04 global compilation ).

*Additional comments:*
We will correct all typos and add comments to the text as suggested by the reviewer.

References:
Lisiecki, L. E., and Raymo, M. E.: A Pliocene-Pleistocene stack of 57 globally distributed benthic d18O records, Paleoceanography, 20, 10.1029/2004PA001071, 2005.
Lourens, L. J., Hilgen, F. J., Laskar, J., Shackleton, N. J., and Wilson, D.: The Neogene Period, in: A Geological Timescale 2004, edited by: Gradstein, F., Ogg, J., and Smith, A., Cambridge University Press, Cambridge University Press, UK, 409-440, 10.1017/ CBO9780511536045.022, 2004.
Zeeden, C., Hilgen, F., Westerhold, T., Lourens, L., Röhl, U., and Bickert, T.: Revised Miocene splice, astronomical tuning and calcareous plankton biochronology of ODP Site 926 between 5 and 14.4 Ma, Palaeogeography, Palaeoclimatology, Palaeoecology, 369, 430-451, http://dx.doi.org/10.1016/j.palaeo.2012.11.009, 2013.
Zeeden, C., Hilgen, F. J., Hüsing, S. K., and Lourens, L. L.: The Miocene astronomical time scale 9–12 Ma: New constraints on tidal dissipation and their implications for paleoclimatic investigations, Paleoceanography, 29, 2014PA002615, 10.1002/2014PA002615, 20142014.

---

## Author Comment (AC2) · 28 Feb 2017

Dear Torsten Bickert,

Thank you very much for your very positive and helpful review.

Please find here the reply to your comments:

*Lines 125ff: Tremendous work that has been done to digitalize the older core photographs, and to correct them for uneven brightness. However, due to wall friction during core penetration, many sections show parabolic bent layers. Is that a problem, when splicing the images of different holes, which might be affected differently by friction? Furthermore, these images are compared to data sets, which have been measured just in the center of the split cores (e.g., magnetic susceptibility using a point sensor, reflectance photospectrometry). Is the offset to a core-wide integration of image data of any importance?*

The parabolic bent layers can sometimes be seen in the core images but they have no effect on the correlation and splicing. In fact, the process of cutting the section images from the core box photo allows the user to exclude the outer edges of the section to mitigate the worst disturbance. Typically, a section image is about 110-115 pixels wide and we cut out the middle 100 pixels. Splicing is done using multiple data sets including those measured at the center of the cores. Generally, there is a small uncertainty involved in the exact position of data acquisition and image recording due to small variations in where the zero point for a measurement is.  The core sections have end caps that due to expansion of sediment on the ship can be bulged a bit.  The images are cut at the end caps in a straight line.  Thus small offsets between data sets at each core are possible, which we estimate in the order of ±1cm, but this will have very minor effect on the correlation, splicing or tuning. There is further discussion of the limits of the depth calibration of the images in the section of the CODD User Guide describing the process.

*Lines 140ff: For readers not that familiar with the splicing procedure, I would suggest to explain in more detail the criteria, how the "master record" is chosen out of the aligned holes. This is in particular important, to understand, why in a second step the sections outside the splice may be stretched and squeezed, instead of being implemented in the master record with their original length.*

This is a very good comment, and we will add a few sentences explaining the splicing criteria in the revised version for clarification. We use the same criteria as typically used by the shipboard stratigraphic correlator for IODP expeditions. The spliced record is composed of core sections from adjacent holes so that coring gaps in one hole are filled with core intervals from an adjacent hole. The splice should contain no coring gaps, and an effort has been made to minimize inclusion of disturbed sections. The choice of tie points (and hence of a splice) is partly a subjective exercise. Normally we followed three rules: Where possible we avoided using the top and bottom ~0.5 m of cores, where disturbance resulting from drilling artifacts (even if not apparent in core logging data) is most likely. We attempted to incorporate those portions of the recovered core that were most representative of the overall stratigraphic section of the site. And we tried to minimize the number of tie points to simplify sampling.

Once the master record where the best quality cores are defined, all intervals outside the splice have to correlated to the master record. Typically these adjustments are in the order of cm to dm mostly due to coring induced compression or stretching of core intervals. In heavily sampled sites it is not uncommon for samples to be taken from core sections that are not part of the splice. The stretching operation ensures that those "off-splice" samples correspond as closely as possible to the same lithology as samples taken from splice sections.

*Lines 215ff: Since the discrepancy between of the interval 1,80 to 1,90 Ma is the largest in the Pleistocene part of the LR04, you should maybe illustrate what might have been the problem for Lisiecki and Raymo (2005) in that interval by exhibiting the original records used (see also suggestions for Fig 10).*

We modified Fig. 8 ( attached ) to better illustrate the mismatch by plotting the reference records used by *Lisiecki and Raymo (2005)* separately.

*Lines 223ff: I fully agree that the tuning of the distinct cyclicity in lithology to orbital precession is robust and of good help as a control for oxygen isotope stratigraphy in the interval between 4.0 to 4.5 Ma. However, again I would prefer to see in separate figure, what might have been the problems of LR04 tuning in that interval, to better follow the arguments presented in the discussion (lines 236ff).*

Figure 10 has been modified to include the LR04 reference records and is attached.

*Fig 7 Abbreviations on 2. y-axes should be explained in the caption. Laskar 2005 => 2004?*

We will correct this typo and add explanation of abbreviations in the revised version!

*Fig 8 Since this figure contains the main results of the study, I would suggest to stretch the two graphs and present them on one page each in a portrait format. Furthermore, the offset of the individual d18O records should be raised to better get access their correlation. Larger data gaps (in particular in sites 927 and 929) should be left open.*

Figure 8 Revised is now in landscape mode to stretch the data. Offsets have been increased. We haven't made this into 2 figures as we feel there is sufficient detail in the redrafted figure.

*Fig 10 I would reorganize this figure in a way that below the results of the Ceara Rise stack, you should probably present at the original data sets of the LR04 stack, to get behind the problem of the former stratigraphy within the interval 1.8 to 1.9 Ma. The lower graph should be moved into a separate figure, and maybe stretched to better present the details of the interval 4.0 to 5.0 Ma.*

We agree with this suggestion and have redrafted the figure accordingly for the revised version.

[Figure]

Figure 8 Revised: Benthic oxygen isotope data from all Ceara Rise sites compared with one another and a smoothed composite of all data compared to LR04. Top - 0 to 2.5 Ma, bottom 2.5 to 5 Ma. Note the $\delta^{18}O$ scale change between top and bottom plots. Individual site traces have been offset as indicated in the legend.

[Figure]

Figure 10 Revised (will be Figure 9 in the revised version after moving Fig. 9 to the supplement): Detail from Figure 8 comparing individual holes to one another and a smoothed composite to LR04. Below the LR04 stack the initial alignment target records from the LR04 stacks are plotted. For the 1.5 to 2.0 Ma interval these are the records from ODP 677 and 849, for the interval 4.0 to 5.0 ma these are the records from ODP 846 and 999. Some records have been shifted as indicated in the figure for better comparison of the data with each other. Note the differences between LR04 and the Ceara Rise average at 1.80 - 1.85 Ma although the initial alignment targets are more similar to the Ceara Rise smooth. Also note the difference between 4.0 and 4.5 Ma. The 999 record is from a single hole and the splice of the 846 record might be erroneous. The age model for the Ceara Rise is very robust in this interval (see. Fig. 11) pointing to potential inconsistencies in the age model construction of the 846 and 999 records.

---

## Author Response (AR1)

Point by point narrative of edits in response to reviewers' major comments:

A list of tables and website addresses for data and code plus Fig S1 ( Fig 9 in original submission ) have been added as a supplementary page.

Line numbers refer to the markup version of the manuscript.

**Zeeden:**

The title of the manuscript has been changed to reflect the 0 to 5 Ma ages considered.

new tables have been added:
        Leg154_Combined_Benthic_Isotopes
        Leg154_Smoothed_Benthic_Isotopes

The MS data are already included in individual site tables. Each data set has the equivalent Site 926 depths included.

Line 110 added web location of data files and software

Line 149 added references on lighting correction by earlier authors.

Line 162 added several sentences addressing the possibility of 9/10 m long lighting variation signals.

Lines 236 & 242 added to cover the relationship between MS and insolation and add a reference to the approach of Zeeden et al.

Line 265 Moved fig 9 to supplement.

Line 286 added paragraph addressing uncertainty.

**Bickert:**

Line 142 answered question regarding distortion of core along outer edges

Line 182 added reference to Ruddiman. We did not want to add more detail within the manuscript itself beyond what we already have since splicing is understood by most of the community and those not familiar may check the reference.

Figures 8 and 10 have been changed to add more data and are now figures 8 and 9.

All of the minor suggestions have been addressed as well.

[revised manuscript text omitted]